



# Multi-star calibration in starphotometry

Liviu Ivănescu[1] and Norman T. O'Neill[1]

[1]Département de géomatique appliquée, Université de Sherbrooke, Sherbrooke, QC, Canada

**Correspondence:** Liviu Ivănescu (Liviu.Ivanescu@usherbrooke.ca)

**Abstract.** We explored the improvement in starphotometry accuracy using a multi-star Langley calibration in lieu of the more traditional one-star Langley approach. Our goal was a 0.01 calibration-constant repeatability accuracy, at an operational sea-level facility such as our Arctic site at Eureka. Multi-star calibration errors were systematically smaller than single star errors and, in mid-spectrum, approached the 0.01 target for an observing period of 2.5 h. Filtering out coarse mode (super $\mu$m) contributions appears mandatory for improvements. Spectral vignetting, likely linked to significant UV/blue spectrum errors at large airmass, may be due to limiting field-of-view and/or sub-optimal telescope collimation. Starphotometer measurements acquired by instruments that have been designed to overcome such effects may improve future star magnitude catalogues and consequently starphotometry accuracy.

## 1 Introduction

Starphotometry involves the measurement of attenuated starlight in semi-transparent atmospheres as a means of extracting the spectral optical depth, and thereby estimating columnar properties of absorbing and scattering constituents such as aerosols, trace gases and optically thin clouds. One of the earliest comprehensive investigations of starphotometry errors and their influence on calibration was reported in the astronomical literature by Young (1974). Calibration strategies for retrieving accurate photometric observations in variable optical depth conditions were proposed by Rufener (1964, 1986). Those studies were recently updated and complemented using starphotometer measurements from our High Arctic, sea-level observatory at Eureka, NU, Canada (Ivănescu (2015), Baibakov et al. (2015), Ivănescu et al. (2021)). This, more recent work, underscored certain challenges in performing calibration at such a high-latitude/low-altitude site. The remoteness of the Eureka site and the significant infrastructure requirements of the starphotometer render calibration campaigns at, say, a dedicated mountain site, onerous. The alternative to a calibration campaign for operational sites (particularly at an Arctic site like Eureka) is to improve on-site calibration methods by overcoming the relatively large optical depth variability at operational sites. Much can be learned by exploring this option at an Arctic location like Eureka(see O'Neill et al. (2016) for a discussion of optical depth variability at Eureka).

Star-dependent (one-star) Langley calibration that depends on large airmass variations is the current standard in starphotometry (see Pérez-Ramírez et al. (2008)Pérez-Ramírez et al. (2011)). This is mainly due to the limited accuracy of available extraterrestrial star magnitudes Ivănescu et al. (2021). A good number of High Arctic stars cannot, however, be so calibrated since they do not go through large elevation (i.e. airmass) changes (in the extreme case of a site at the pole, there are no eleva-





tion changes). Our goal is to demonstrate that a sub 0.01 optical depth error (partly linked to calibration errors) can be achieved by performing the type of instrument-dependent, star-independent calibration referred to in Ivănescu et al. (2021).

## 2 Calibration methodology

### 2.1 Langley calibration

The starphotometer retrieval algorithm is based on extraterrestrial and atmospherically attenuated magnitudes of non-variable bright stars, denoted by $M_0$ (usually provided by a catalog) and $M$, respectively (see Ivănescu et al. (2021) for the nomenclature details). Their corresponding (instrument) signals, expressed in terms of a magnitude (logarithmic) formulation, are $S_0$ and $S$, respectively. The star-independent conversion factor between the catalog and instrument magnitudes is (ibid)

$$C = M - S \tag{1}$$

$$C = M_0 - S_0 \tag{2}$$

The $C$ factor accounts for the optical and electronic throughput of the starphotometer, as well as the photometric system transformation between the instrument signal magnitude and the extraterrestrial catalog magnitude. In terms of magnitude, the Beer-Bouguer-Lambert atmospheric attenuation law is

$$M = M_0 + (m/0.921)\tau \tag{3}$$

where $m$ is the observed airmass and $\tau$ is the total optical depth. Inserting equation (1) yields

$$M_0 - S = -\tau x + C \tag{4}$$

where $x = m/0.921$. This expression can be used to retrieve $C$ from a linear regression of $M_0 - S$ versus $x$, if $\tau$ is assumed constant. Such a procedure is referred to as the Langley calibration technique, or Langley plot. In the absence of an accurate $M_0$ spectrum, equation (2) can be used to transform equation (4) into

$$S = \tau x + S_0 \tag{5}$$

for which a catalog is no longer required. This linear regression enables the retrieval of $S_0$ instead of $C$ and thus represents a star-dependent calibration.

The right side of equation (4) notably indicates that $M_0 - S$ is star independent: it thus represents a linear regression that any star can contribute to and, accordingly, a framework for multi-star Langley calibration.



## 3 Calibration errors

### 3.1 Measurement accuracy

The differential of (rearranged) equation (4) yields the calibration accuracy error

$$\delta_C = (\delta_x \tau + x \delta_\tau) - \delta_S + \delta_{M_0} \tag{6}$$

The $(\delta_x \tau + x \delta_\tau)$ component underscores the rationale for performing calibrations at a high altitude site (where $\tau$, $\delta_\tau$ and $\delta_x \tau$ are typically smaller) and the advantage of maintaining small $x$ in order to minimize the $x \delta_\tau$ contribution to $\delta_C$. The sky stability during the retrieval of $C$ may be monitored by computing $\tau$ for each sample, with equation (4). The $\delta_S$ error component accounts for any systematic signal changes: optical transmission degradation, misalignment error, star spot vignetting etc. The $\delta_{M_0}$ component accounts for any magnitude bias in the bright star catalog (i.e., it is the average of accuracy-error spectra

for all catalog stars: see Ivănescu et al. (2021) for a detailed discussion of error bias in the catalog stars). Because it is a catalog-specific constant, the optical depth accuracy will not be affected by its consistent use[1].

### 3.2 Regression precision

A linear regression applied to a plot of $y = M_0 - S$ versus $x$ yields the slope $(-\hat{\tau})$ and intercept $(\hat{C})$ of the Langley equation (4). The regression equation is then $\hat{y} = -\hat{\tau} x + \hat{C}$ and the linear-fit residuals are represented by $r = y - \hat{y}$. The standard error of

the regression slope and intercept for a large number of measurements[2] can be expressed as (see, for example, Montgomery and Runger (2011))

$$\sigma_{\hat{\tau}} = \frac{\sigma_{\bar{r}}}{\sigma_x}, \qquad \sigma_{\hat{C}} = \sigma_{\hat{\tau}} \sqrt{\overline{x^2}} \tag{7}$$

It should be noted that $\bar{r}$ (the mean of the residual) $= 0$ is a corollary of the linear regression constraints.

The Langley calibration y-axis embodies two independent sets of measurements: $N$ "measurements" of $M_0$ and $n$ mea-

surements of $S$. From a pure noise standpoint, the residuals can be represented by an ensemble of individual measurements $(r = (M_0 - S) - (-\tau x + C))$ where each parameter (except $C$) is subject to noisy variation. Excluding the typically negligible random errors in $x$ yields[3]

$$\sigma_{\bar{r}}^2 = \frac{\sigma_{\epsilon_S}^2}{n} + \frac{\sigma_{\epsilon_\tau}^2 \overline{x^2}}{n} + \frac{\sigma_{\epsilon_{M_0}}^2}{N} = \sigma_{\bar{\epsilon}_S}^2 + \sigma_{\bar{\tau}}^2 \overline{x^2} + \sigma_{\bar{\epsilon}_{M_0}}^2 \tag{8}$$

where the standard error expression for a linear combination of random variables was employed (Barford (1985)). The subscript

$\epsilon$ represents a single instance of a random (noise) measurement in $S$, $\tau$ or $M_0$ and $\sigma_\epsilon$ is its zero-mean standard deviation. $\sigma_{\epsilon_\tau}$ was replaced by $\sigma_\tau$ because no systematic variation was assumed in $\tau$ during the calibration period. $\epsilon_{M_0}$ represents the

---

[1]Such an error becomes part of the $C$ value extracted from the Langley calibration of equation (4) and becomes part of the operational retrieval process when equation (4) is inverted to yield individual values of $\tau$.

[2]$n > 10$ where $n = \sum n_i$ ($n_i$ being the number of observations associated with star i)

[3]Where the variance of the $\epsilon_\tau x$ product (Goodman (1960)) is $\sigma_{\epsilon_\tau x}^2 = \sigma_x^2 \bar{\epsilon}_\tau^2 + \sigma_{\epsilon_\tau}^2 \overline{x^2} = \sigma_\tau^2 \overline{x^2}$, since $\bar{\epsilon}_\tau = 0$ and $\sigma_{\epsilon_\tau} = \sigma_\tau$.





difference between an individual star's $M_0$ accuracy-errors and the averaged $M_0$ catalog bias. The $\sigma_{\epsilon_{M_0}}$ term is specific for the use of multiple stars during the calibration.

## 4  Observing conditions

The assumption of constant $\tau$ in time ($t$) and observational direction (expressed in terms of $m$) may be problematic over long observation periods and large airmass changes. It is a useful exercise to assess the average time period and airmass range over which a degree of $\tau$ constancy (sky stability) is maintained.

Variations of a sky instability parameter ($\sigma_{\delta\tau}$) were analyzed using $\delta\tau$ differences for $\tau$ measurements acquired during the 2019-2020 season in Eureka. $\delta\tau$ values were placed into (a) fixed $\Delta t$ bins to generate $\delta\tau$ histograms for high stars (where

$\delta\tau = \tau_f - \tau_i$ is computed from a later time (f) relative to an earlier time (i), and (b) fixed $\Delta m$ bins from high- to low-star $m$ pairs. Since $\delta$ of each bin generally come from distinct periods, $\tau_f$ and $\tau_i$ are expected to be uncorrelated: the $\tau_i$ versus $\tau_f$ correlation coefficient was determined to be $< 0.25$ when $\tau_p < 0.1$, $\Delta t < 1\,$h and $\sim 0.1$ otherwise (see the legend of Figure 1 for the definition of $\tau_p$). This is negligible for the purposes of our analysis and, accordingly, they can be considered as independent variables. The approximation $\sigma_\tau \cong \sigma_{\delta\tau}/\sqrt{2}$ (Soch et al. (2021)) can accordingly be employed for each $\Delta m$ or $\Delta t$ bin.

Those histograms often included anisotropic outliers typical of lognormal $\tau$ statistics (Sayer and Knobelspiesse (2019)). A median approach was chosen to render the statistics approximately independent of the outliers: the MAD (Median Absolute Deviation) parameter was employed as a robust measure of histogram width (see equation (1.3) in Rousseeuw and Croux (1993) for MAD details). In order to eventually convert the statistics to those of a normal distribution, an outlier cutoff of $4.5 \cdot$ MAD was defined[4]. This particular cutoff is equivalent to the classical normal distribution outlier cutoff of $3\sigma$ since $\sigma = 1.5 \cdot$ MAD.

Figure 1 shows $\sigma_\tau$ (computed after the outlier cut-off and using the $\sigma_\tau$ approximation given above) as a function of (a) $\Delta t$ and (b) $\Delta m$. It can be shown[5] that a calibration period of 2 h, for which $n \simeq 46$ at the standard sampling rate of starphotometer, yields $\sigma_\tau \simeq 1.4\sigma_{\hat{C}}$. This means that the calibration error ($\sigma_{\hat{C}}$) is limited to $< 0.01$ only if $\sigma_\tau < 0.014$. An 8-h observing period enables a more generous limit of $\sigma_\tau < 0.028$ to achieve the same calibration precision. Contour curves of $\sigma_\tau = 0.014$ and $0.028$ are superimposed on Figure 1.

Figure 2 shows the $\sigma_\tau$ variability estimation for the 2-h "fast" and the 8-h "long" calibration periods, as well as a third scenario with $\Delta m = 1$ to 5. The three curves represent the standard deviation (after cut-off) of the corresponding range-aggregated data. They tend to converge with decreasing $\tau_p$: the 2-h and 8-h $\sigma_\tau$ values of $0.014$ and $0.028$ correspond to $\tau_p$ values of $0.13$ and $0.15$, respectively (blue and red dashed vertical lines defined by the intersection with the corresponding horizontal $0.014$ and $0.028$ lines). The cases $\tau_p \leq 0.13$ and $0.15$ were labeled as "clear-sky" conditions because of their tendency to

promote calibration stability. Their corresponding clear-sky statistics are presented in appendix A.

---

[4]A cutoff liberty that we availed ourselves of because one is free to chose the duration of the calibration period and/or to perform outlier filtering prior to Langley regressions.

[5]Using $\sigma_\tau^2 \simeq \sigma_{\hat{C}}^2 n/k_3$ (obtained from equation (B3)), with the terms in $S$ and $M_0$ neglected, and inserting $\sigma_\tau/\sqrt{n}$ (i.e. $\sigma_{\overline{\tau}}$) into equation (7) and noting that a typical range of $x \in [1.086, 5]$ yields $k_3 \simeq 23$ (see Figure B1).

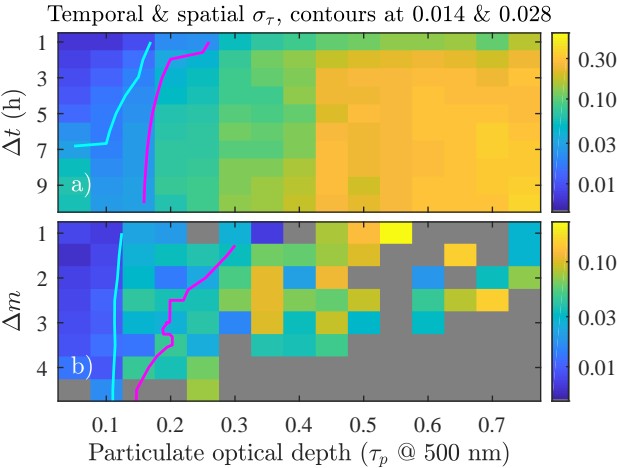

**Figure 1.** Two dimensional sky instability ($\sigma_\tau$) patterns for the 2019-2020 season at Eureka. The colour coded $\sigma_\tau$ values are computed relative to a reference $\tau$ value but plotted as a function of its associated particulate optical depth ($\tau_p = \tau - \tau_m$, where $\tau_m$ is the molecular scattering optical depth) and a) time difference ($\Delta t$), or b) air mass difference ($\Delta m$). The magenta and purple curves represent the column-wise averaged $\sigma_\tau = 0.014$ and 0.028 contour lines. There was much more data associated with $\Delta t$ than with $\Delta m$ bins (i.e. more robust bin statistics are expected in the former case). Note that $\Delta m$ and $\Delta t$ were chosen to be positive.

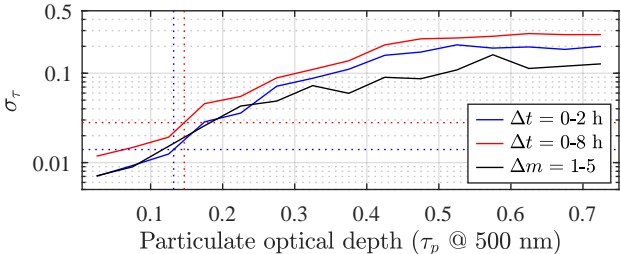

**Figure 2.** $\sigma_\tau$ *vs* $\tau_p$ for the three calibration scenarios defined.

Many high Arctic stars are circumpolar (i.e. they never set) and thus their airmass range is limited. Figure 3 shows airmass variation as a function of time past the transit for our dataset of the 13 brightest (and stable) stars at Eureka. A well defined separation is notable between high stars ($m(12h) < 3.1$) and low stars ($m(0h) > 2.2$). A large airmass range is clearly only available for the low stars (i.e. about 2/3 of our Eureka bright-star dataset). However, star vignetting, due to turbulence-inducing star-spot expansion beyond the boundaries of the Field of View (FOV), may affect the optical throughput of the Eureka system at $m > 5$ (Ivănescu et al. (2021)). This type of airmass constraint, combined with the low star constraints of Figure 3, results in only moderate $\Delta m$ excursions (at the expense of substantial $\Delta t$) if only a single star is employed in a Langley type calibration. A multi-star calibration can be exploited to mitigate such $\Delta m$ and $\Delta t$ limitations.

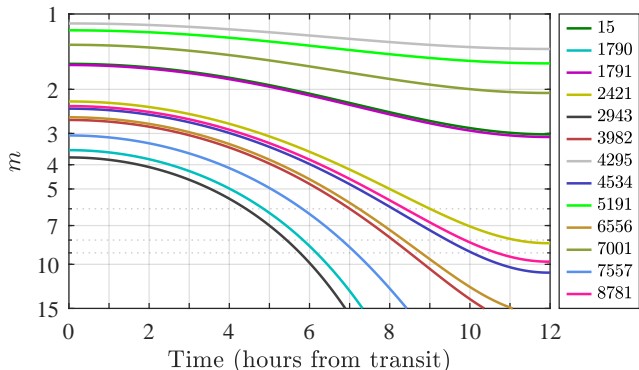

**Figure 3.** Airmass versus time past the transit for the bright stars observable at Eureka (identified by their HR catalog index). The transit of a given star occurs when it crosses the local meridian (minimum airmass).

## 5 Multi-star calibration

This type of calibration exploits a singular advantage of starphotometry over moonphotometry and sunphotometry: the capability of employing multiple extraterrestrial light sources in a relatively short period of time. In comparison with a C-determining Langley calibration using one star, the multi-star approach enables a synergistic Langley calibration that employs several stars exhibiting a wide range of airmass values over a significantly shorter period of time.

One- and multi-star Langley calibrations acquired with the Eureka starphotometer on 2019/12/07 and 2020/01/10, respec-
tively, are shown in Figure 4. The observations for $x > 5$ were carried out to highlight any vignetting effect due to the aforementioned star-spot expansion. The one-star case (small black dots and their associated "1-lin" regression line) are the results for the low Procyon star (HR 2943, spectral type F5V). Its colder temperature ensures a near infrared (NIR) brightness that is larger than all the other bright stars of Figure 3[6]. That reason aside, it is also, arguably, the most optimal one-star Langley-regression choice since no other Figure 3 bright star can duplicate its large and rapid airmass change (c.f. the lowest black
curve).

### 5.1 Calibration precision

The resulting one- and multi-star $\hat{\tau}$ spectra (each spectral point representing a linear-regression Langley slope) are shown in Figure 5a. Their associated precision errors ($\sigma_{\hat{\tau}}$) of equation (7) are shown in Figure 5b. One should note that the estimated multi-star error is substantially and consistently smaller than that of the one-star calibration. The $\hat{C}$ and $\sigma_{\hat{C}}$ spectra from the
Langley regressions are shown in Figure 6a and Figure 6b, respectively. The $\sigma_{\hat{C}}$ values are, in the multi-star case, significantly smaller and closer to the 0.01 target.

The generally smaller $\sigma_{\hat{\tau}}$ values of the multi-star case are partly attributable to the one-star case being limited to a relatively smaller $x$ range (i.e. smaller $\sigma_x$ in equation (7)) while the smaller $\sigma_{\hat{C}}$ values are partly attributable to the smaller $\sigma_{\hat{\tau}}$ values

---

[6]The other bright stars, being of similar A-B type (*ibid*), generally exhibit lower signal-to-noise (SNR) in the NIR.



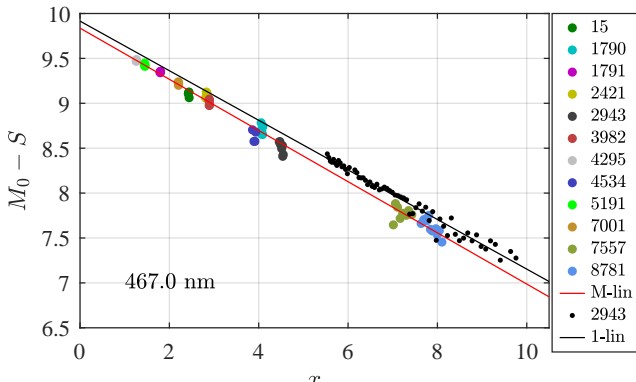

**Figure 4.** One- and multi-star Langley calibrations (both lasted about 2.5 h, see appendix A). The one-star and multi-star measurement points are represented, respectively, by small black dots and large solid-colored circles, while their linear regression fits appear as solid lines (1-lin and M-lin, respectively). Each point represents an average of five 6 s exposures. Each star is identified by their HR IDs (Ivănescu et al. (2021)).

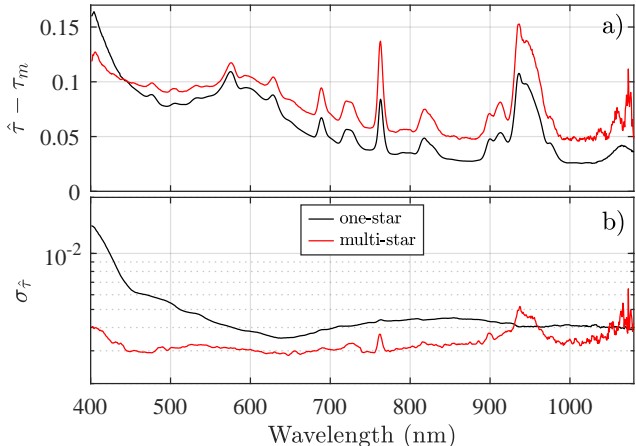

**Figure 5.** a) One- and multi-star Langley-regression slopes (expressed as $\hat{\tau_p} = \hat{\tau} - \tau_m$). b) $\sigma_{\hat{\tau}}$ values derived from eq. (7).

and the generally lower values of $x$ (c.f. equation (7b)). The $\sigma_{\hat{C}}$ increases in the ultraviolet (UV) and NIR are discussed in
subsection 6.2. The peak around 940 nm is likely associated with a faint and noisy star signal induced by strong attenuation in the water vapour absorption band, coupled with the non-linear nature of the optical depth in that spectral region (Pérez-Ramírez et al. (2012)).



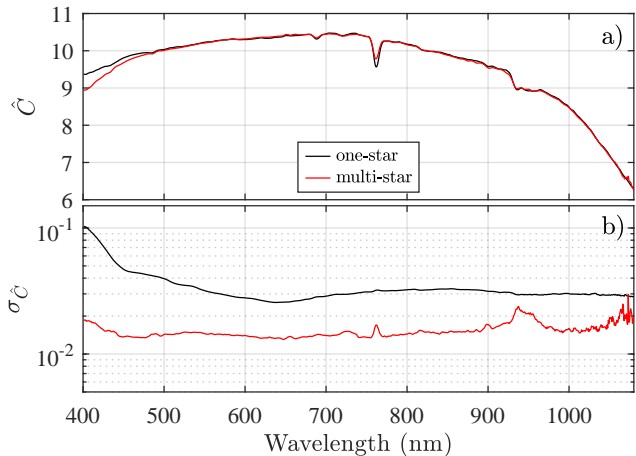

**Figure 6.** a) $\hat{C}$ retrieved from the one- and multi-star Langley calibrations. b) $\sigma_{\hat{C}}$ values computed using equation (7).

## 5.2 Repeatability

The robustness of the $\sigma_{\hat{C}}$ spectra of Figure 6b and the impact of potential systematic errors can be investigated with repeatability
experiments. The $\hat{C}$ spectra employed to produce the standard deviations[7] shown in Figure 7 were derived from three one-star
and three multi-star Langley calibrations that were well separated in time (i.e., they were optically independent in terms of any
significant correlations between the $\tau_p$ variations of each period) and nearly satisfied the clear-sky calibration constraints of
section 4. The Figure 7 error spectra are, with the exception of larger differences in certain spectral regions, roughly coherent
with the Figure 6b spectra (including the fact that the one-star errors are significantly larger than the multi-star errors).

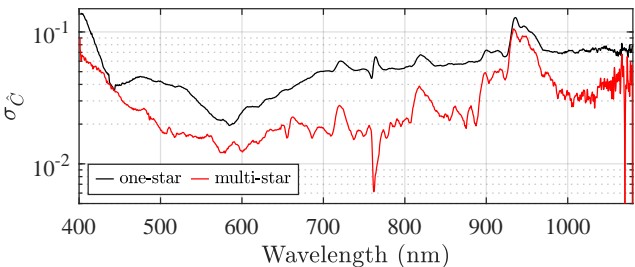

**Figure 7.** $\sigma_{\hat{C}}$ curves derived for three one-star Langley calibrations acquired using the Procyon (HR 2943) star on 2019/12/07, 2020/01/05 and 2020/01/16, as well as three multi-star calibrations acquired on 2018/03/10, 2019/12/07 and 2020/01/10. These spectra are generally similar to Figure 6b results.

---

[7]Standard deviations that, we would argue, are also standard errors (each of the three $\hat{C}$ spectra that were averaged were more akin to means).




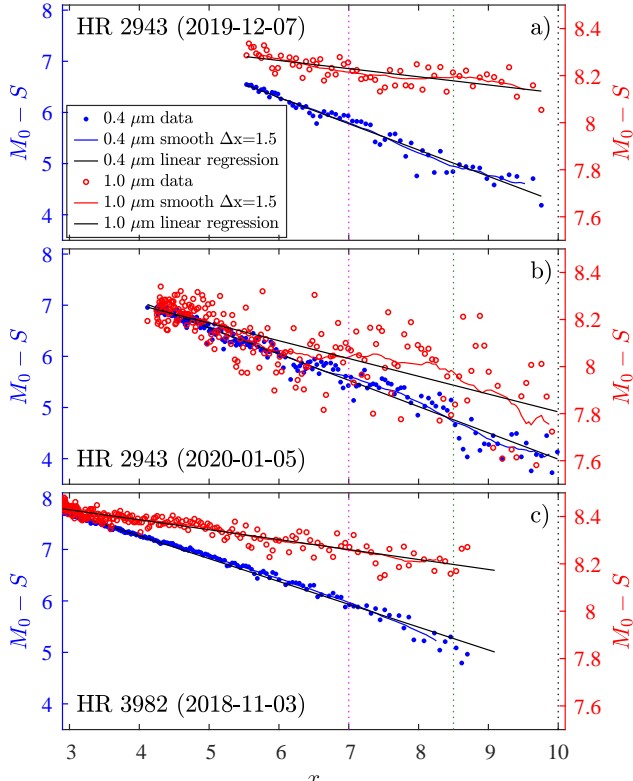

**Figure 8.** One-star calibrations at two spectrally distinct channels (400 and 1000 nm) for different dates and stars. The black lines represent the full regression line for all data points over the entire $x$ range. Star measurements for the a) and b) cases started at the smallest $x$ values, while the c) case measurements began at the largest $x$ value. The solid (varying) curves were generated by averaging $(M_0 - S)$ over $\Delta x = 1.5$ sliding windows. The three colored (dotted) vertical lines correspond to the colors of the three $x_{max}$ cases of Figure 10.

## 6   Regression error discussion

### 6.1   Data processing

Figure 8 shows dual wavelength (400 and 1000 nm) regression tests for two of the three one-star calibrations of the previous section (two of the three dates given in the legend of Figure 7 for the HR 2943 star) plus a third hotter star (HR 3982, spectral type B7) that was specifically chosen to better understand the influence of temperature-driven spectral differences in the target star. The smaller regression-slope and point dispersion about the HR 3982 regression line, compared with the two HR 2943 cases, is noticeable at both wavelengths (notably at 1000 nm) and is an indicator of generally clearer sky conditions.

The $C$ values retrieved from linear regressions over an increasing $x$ range in Figure 8 (from the smallest $x$ value to an artificial maximum of $x_{max}$) are plotted in Figure 9. The damping out of regression noise and the asymptotic approach to the horizontal pan-$x$ regression value as $x_{max}$ increases can be readily observed in all three plots.





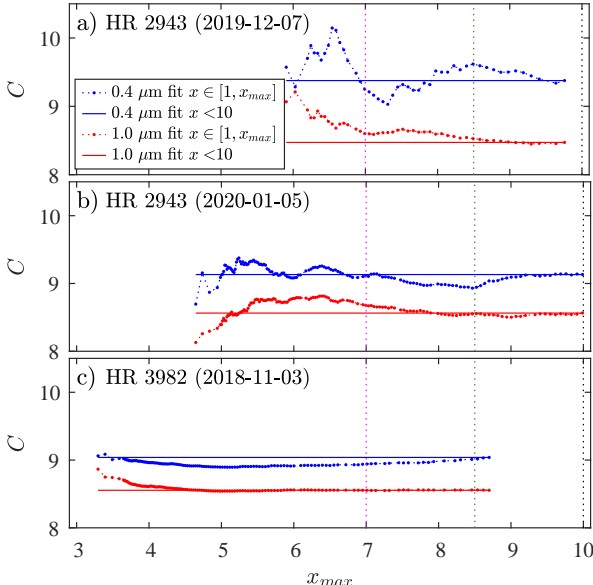

**Figure 9.** $C$ values retrieved from linear regressions over an increasing $x$ range, i.e. from the smallest $x$ to an increasing $x_{max}$ for all the cases plotted in Figure 8. The horizontal reference lines represent regressions over the entire $x$ range (the solid lines of Figure 8), while the three colored (dotted) vertical lines correspond to the colors of the three $x_{max}$ cases of Figure 10.

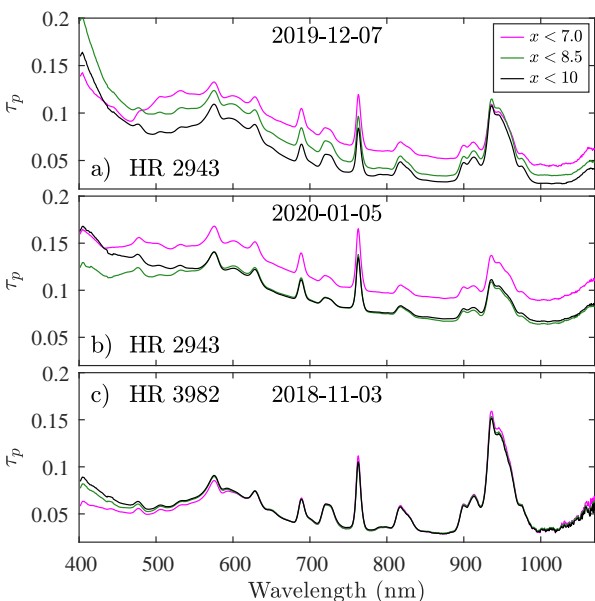

**Figure 10.** Optical depth (slope) spectrum retrieved from calibration performed at different $x$ ranges.



The corresponding slope-derived $\tau_p$ spectra are shown in Figure 10 for three $x_{max}$ cases (the three colored spectra were derived for $x_{max}$ values corresponding to the matching colors of the three vertical lines in Figure 8 and Figure 9). The $x$-dependent regression error dynamics are investigated in appendix C. The next subsection describes potential competing causes of $C$ variations and makes a link to $\tau_p$ errors[8].

## 6.2   Regression error interpretation

The sky instability plots of Figure 1 show that the standard deviation of the optical depth increases with time and airmass separation between any two stars (this applies equally well to the variation between two positions of the same star). A systematic optical depth drift during the calibration leads to a common-signed bias (positive or negative) of the regression slope and the calibration value, relative to drift free conditions. Figure 10a and 10b show spectrum-wide $\tau_p$ reduction as $x_{max}$ and calibration time increase. This suggests spatial and/or temporal sky transparency instability during calibration. Such rapid and spectrally

neutral variation is consistent with the domination of coarse mode (super $\mu$m) particles: a (post cloud-screened) mode that is mostly dominated by spatially homogeneous cloud particles at Eureka (O'Neill et al. (2016)). The near-superposition of all $\tau_p$ spectra above 500 nm in Figure 10c indicates stable transparency that is characteristic of a cloud-free atmosphere dominated by fine-mode (sub $\mu$m) particles. A number density induced drift of similar fine-mode aerosol particles will generate spectrally independent variations in $\Delta\tau_p/\tau_p$: the larger $\tau_p$ value (corresponding to the larger absolute difference in the blue/UV part of

the spectrum) could explain the increasingly larger UV deviations (such as between the magenta and black/green curves in Figure 10c).

    The two bullet-lists below summarize the specific processes that can lead to variations of calibration slope ($\tau_p$) and intercept ($C$), traceable to real or apparent optical depth variations.

*Instances of $\tau_p$ and $C$ overestimation*

• A systematic coarse-mode $\tau_p$ increase (as described above) can have a dramatic spectrum-wide effect: flagging and discarding such measurements is, accordingly, essential. A fine-mode $\tau_p$ increase will predominantly affect the UV/blue part of the spectrum.

    • Recent tests indicate that the optical collimation of the Eureka Celestron C11 telescope requires correction. Mis-collimation is responsible for a significant part of the star spot size reported in Ivănescu et al. (2021). Correcting the attendant vi-

gnetting problem (whose consequence is a decreased star flux and apparent increase in $\tau_p$) may enable reliable measurements at $x$ values well above the limit of $x \simeq 5$ reported by Ivănescu et al. (2021).

    • The angular star spot size ($\omega$), being proportional to $\lambda^{-1/5}x^{3/5}$ (equations 4.24, 4.25 and 7.70" of Roddier (1981)), effectively leads to spectrally dependent vignetting (i.e. apparent $\tau_p$ and $C$ increase) as a function of $x$: an increase in $x$ from 7 to 9.5 would be equivalent to 20% increase for a spectral change from 400 to 1000 nm. This coupled spectral

and airmass vignetting influence is consistent with Figure C1 with the blue (0.4 $\mu$m) curve increasing at $x \simeq 7$ while the

---

[8]The strong, positive correlation between $C$ and $\tau_p$ and between their errors is the result of variations in the regression lines being effectively driven by rotations about a cluster of pivot points whose x position changes little.




increase of the red (1.0 $\mu$m) curve occurs only at $x > 9$. This dynamic potentially dominates the large UV/blue errors seen in Figure 7 and Figure 10.

- Noisier star spots, attributable to increased turbulence and scintillation at large $x$, may induce larger centering errors and exacerbate apparent increases in $\tau_p$ and $C$ due to vignetting.

*Instances of $\tau_p$ and $C$ underestimation*

- A systematic $\tau_p$ decrease during the calibration period (notably when the calibration starts at large $\tau_p$).
- Weak signals, usually at large $x$ and notably for hot stars in the NIR, may lead to sensitivity loss due to ADC (analog to digital conversion) limitations and attendant slope and intercept ($\tau_p$ and $C$) reductions.

These factors contribute to Figure 10 $\tau_p$ dynamics and likely relate to the one-star $\sigma_{\hat{C}}$ spectra shown in Figure 7. A very
similar spectrum is indeed observed in the case of one faint star at large airmass (Figure 11). Such spectral dynamics, possibly dominated by the aforementioned spectral influence of vignetting, are also likely related to the similar $M_0$ bias spectra shown in Figures 4 and 11 of Ivănescu et al. (2021). The identification of the $M_0$ bias source is of paramount importance, as it may guide strategical observation choices made to improve the accuracy of future star catalogues. The error envelopes about the $M_0$ bias (quantified in Figure 12) add an additional, roughly flat spectral component (in spectral regions other than those those that are dominated by H-absorption bands).

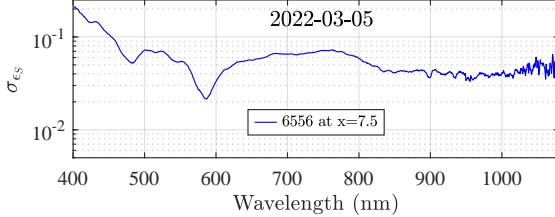

**Figure 11.** Standard deviation of $S$ magnitude measurements at large airmass for a faint catalogue star (HR 6556, V=2.08).

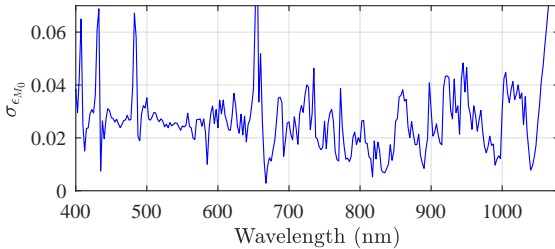

**Figure 12.** Standard deviation of $M_0$ errors deduced from the error bars of Figure 4 in Ivănescu et al. (2021).


The smoother NIR errors in the one-star case (comparing the black one-star curve with the red multi-star curve of Figure 5a for $\lambda > 1050$ nm) is likely due to the strong NIR signal of the much colder Procyon star. One can take advantage of this effect



and develop an observing strategy that avoids using faint stars at large airmass at Eureka and still employ 12 catalogue stars at $x<8$ in a multi-star calibration lasting 2.5 h (c.f. Figure A2). The star selection operation for a given multi-star calibration

should also include a random airmass selection to mitigate accuracy errors attributable to systematic optical depth variations (as an alternative to the Rufener (1986) method). Mitigation of both, starlight reduction impacts at large airmass and systematic optical depth variations is a singular advantage of the multi-star vs one-star calibration.

## 7    Conclusions

It was determined that no Eureka star movement satisfied an optimal sky-transit scenario of maximum possible airmass range

within the constraint of $x$ being $\lesssim 5$. The solution to this intrinsic shortcoming of a High Arctic site is to perform multi-star calibrations: this approach incorporates the fundamental advantage of reducing the calibration period and thus minimizing optical depth variability. It is, by its very nature, a calibration that enables the retrieval of a star-independent calibration parameter.

Multi-star calibration repeatability errors ($\sigma_{\hat{C}}$) were systematically smaller than the single star errors and, in the central part of the spectrum, approached the target value of 0.01 for an observing period of 2.5 h. Those errors were partly affected by

less than optimal clear sky conditions (notably in the presence of cloud), with $\tau_p$ larger than the recommended "clear-sky" value of 0.13: c.f. section 4 and appendix A). Coarse mode filtering algorithms, that ideally eliminate all influences of coarse mode optical depth specifically in a calibration scenario, are necessary to ensure the best calibration[9]. Large UV and NIR errors can be reduced by avoiding faint stars at large $x$ and by improving the current telescope collimation. The mitigation of mis-collimation problems can, in the short term, be affected by a constraint of $x < 7$. This can be achieved at Eureka by

employing 12 constrained-magnitude stars over a 3 h calibration period (c.f. appendix A). A constraint of $\tau_p<0.13$ may bring the calibration errors in the blue-to-red spectral range closer to the 0.01 target, with the remaining UV and NIR spectral regions being subject to the influence of $M_0$ errors.

In summary, the advantages of multi-star versus one-star calibration, are star-independent calibration, faster coverage of larger airmass ranges, more calibration opportunities and star selection capability for both mitigating the impact of starlight

reduction with increasing airmass and systematic optical depth variations. These singular benefits were shown to override the drawbacks of specific star catalogue errors (i.e. the multi-star calibration performs better than the one-star case, even if the former is uniquely affected by $M_0$ errors). Further improvement will only be achieved by developing a more accurate extraterrestrial star-magnitude catalog: their UV/blue errors, likely linked to large-$x$ spectral vignetting, are endemic to current ground-based star catalogues. This improvement may be affected from a space-borne platform or at a high-elevation observa-

tory (the primary goal being to reduce turbulence-induced star-spot size and optical depth variability). The use a large aperture telescope (limiting scintillation and low-starlight measurement errors) and a larger FOV instrument (less prone to vignetting) will, in general, provide better results.

---

[9]Clouds are usually the dominant coarse mode component but coarse mode aerosols can have diverse effects which are typically but not always minor.



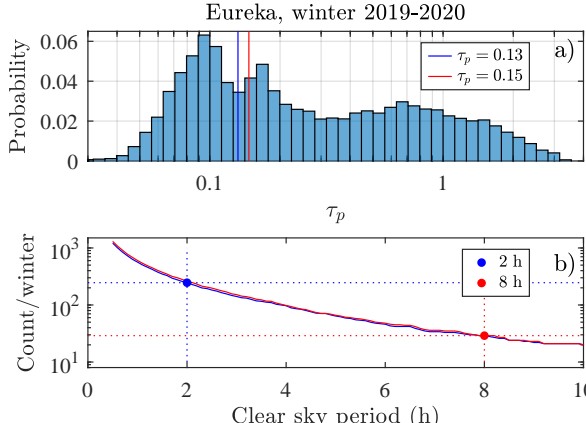

**Figure A1.** $\tau_p$ histogram for measurements acquired during the 2019-2020 observing season at Eureka (total of 25914 measurements). The blue and red vertical lines are the clear-sky cutoff values of 0.13 and 0.15 determined in section 4. The probability is normalized so that its sum is unity (a). Frequency of occurrence vs duration of clear-sky periods (b).

*Code and data availability.* Final MATLAB code and data employed in the generation of the figures is freely available (see Ivănescu (2023))

## Appendix A: Calibration opportunities

Figure A1a shows the $\tau_p$ histogram for data acquired during the 2019-2020 observing season at Eureka[10]. The blue and red vertical lines respectively indicate the clear-sky cutoff values of 0.13 and 0.15 determined in section 4 for the 2-h and 8-h calibrations. Operational conditions occurred 37% of the time (i.e. those periods of time when measurements were not impeded by persistent thick clouds or the performance of maintenance tasks). A frequency curve of clear-sky periods (a period for which all $\tau_p$ values are less than the cutoff value) is presented in Figure A1b. Measurements acquired during 2-h and 8-h clear-sky

periods represented, respectively, 35.5% and 39% of all measurements. These numbers, transformed into an estimation of clear-sky fraction of the total measurement time, yield values of 13% and 14% of the total contiguous seasonal time ($0.37 \cdot 0.355$ and $0.37 \cdot 0.39$, respectively). Since the measurement season is ~160 days (or ~5.3 months[11]) and given that there were 246 clear-sky periods of 2 h with $\tau_p < 0.13$, one may expect 46 such calibration periods per month. There were, on the other hand, 29 clear sky periods of 8 h with $\tau_p < 0.15$ (or ~5.5 per month). If a calibration can be successfully completed in ~2 h then there

is a significantly larger probability-of-occurrence incentive for doing so.

The weakening of star signals with increasing airmass will progressively impact calibration quality. Figure A2 shows the availability of catalogue stars for a multi-star calibration over Eureka as a function of calibration period and maximum airmass.

---

[10]We could speculate that the two histogram peaks near $\tau_p$ values of 0.1 and 0.16 are associated with the background fine mode optical depth and the enhanced fine mode optical depth incited by the presence of wind blown seasalt (O'Neill et al. (2016))

[11]Which we pragmatically define as the number of nights for which reliable measurements can be carried out for $\geq 30$ minutes.



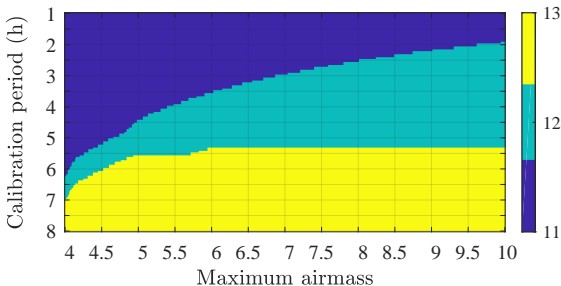

**Figure A2.** Number of available catalogue stars for a multi-star calibration, during a 24 h period. The following constraints were employed in generating the tri-color contours: at least one star at $x < 1.2$, exclusion of any star of visual magnitude $V > 1.5$ for $x > 6$, as well as $V > 2$ at $x > 5$.

A calibration can, for example, be carried out in 2 h with only 11 stars of our 13 star dataset (Figure 3). A 12 star calibration can be carried out only if $x < 9.5$, or if the calibration period is $> 2.5$ h.

**Appendix B: Relative importance of component errors**

From equation (7) and (8) one gets the error propagation into the $\tau$ Langley retrieval

$$\sigma_{\hat{\tau}}^2 = \frac{\sigma_{\bar{\tau}}^2}{\sigma_x^2} = \frac{1}{\sigma_x^2}\left(\sigma_{\bar{\epsilon}_S}^2 + \overline{x^2}\sigma_{\bar{\tau}}^2 + \sigma_{\bar{\epsilon}_{M_0}}^2\right) \tag{B1}$$

$$\sigma_{\hat{\tau}}^2 = k_1\sigma_{\bar{\epsilon}_S}^2 + k_2\sigma_{\bar{\tau}}^2 + k_1\sigma_{\bar{\epsilon}_{M_0}}^2, \text{with } k_1 = \frac{1}{\sigma_x^2}, k_2 = \frac{\overline{x^2}}{\sigma_x^2} \tag{B2}$$

Error propagation into the calibration constant ($C$) retrieval is, in a similar fashion, expressed as

$$\sigma_{\hat{C}}^2 = \sigma_{\hat{\tau}}^2\overline{x^2} = \frac{\overline{x^2}}{\sigma_x^2}\left(\sigma_{\bar{\epsilon}_S}^2 + \overline{x^2}\sigma_{\bar{\tau}}^2 + \sigma_{\bar{\epsilon}_{M_0}}^2\right) \tag{B3}$$

$$\sigma_{\hat{C}}^2 = k_2\sigma_{\bar{\epsilon}_S}^2 + k_3\sigma_{\bar{\tau}}^2 + k_2\sigma_{\bar{\epsilon}_{M_0}}^2, \text{with } k_3 = \frac{\overline{x^2}^2}{\sigma_x^2} \tag{B4}$$

The coefficients $k_1$, $k_2$ and $k_3$ are displayed in Figure B1b, c and d, respectively, for the $x$ protocols of Figure B1a. The blue curve shows uniformly distributed values of $x$, while the red curve shows a more realistic observing configuration of constant time intervals[12]. In order to investigate more practical (smaller) ranges, the working range is incrementally truncated from both,

the right and left (the solid and dashed curves, respectively). The focus is on two particular ranges: $x < 5$ (red-filled circles), where $k_1 \simeq 1.2$, $k_2 \simeq 5.3$ and $k_3 \simeq 23$ (approximately-stabilized for $X \gtrsim 4$), and $x > 5$ (open red circles), where $k_1 \simeq 0.5$, $k_2 \simeq 25$ and $k_3 \simeq 1250$ (i.e. $> 50$ times greater than that for $x < 5$). This strong weighting towards large $x$ drives the standard error in $C$. The term $\sigma_{\bar{\epsilon}_S}$ is typically $\sim \sigma_{\bar{\tau}}$, which is, in turn $\sim \sigma_{\bar{\epsilon}_{M_0}}$ Ivănescu et al. (2021). The $\sigma_{\hat{\tau}}^2$ term may, accordingly, tend to dominate the $\overline{\epsilon_S}^2$ and $\sigma_{\bar{\epsilon}_{M_0}}^2$ terms in equation (B3)[13] and thus the $\sigma_{\hat{C}}$ calibration error.

---

[12]Both conditions apply to a star crossing the meridian at zenith.

[13]Since, as per Figure B1e, $4 < \overline{x^2} < 50$.

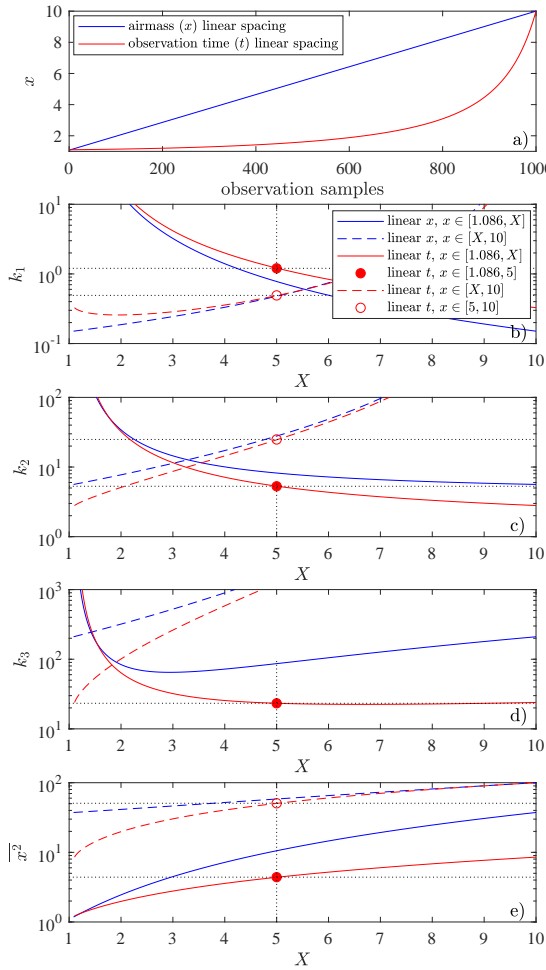

**Figure B1.** Variation of $x$ as a function of the number of observation samples for measurements made in equal increments of $x$ (blue curve) and equal increments of time (red curve) (a). The next three panels show the $x$ dependent variation of $k_1$, $k_2$ and $k_3$ (see text for more details). The legend in panel (b) applies to all the subsequent panels. Panel (e) shows $\overline{x^2}$, the $\sigma_{\hat{\tau}}^2$ to $\sigma_{\hat{C}}^2$ conversion factor of equation (B3).





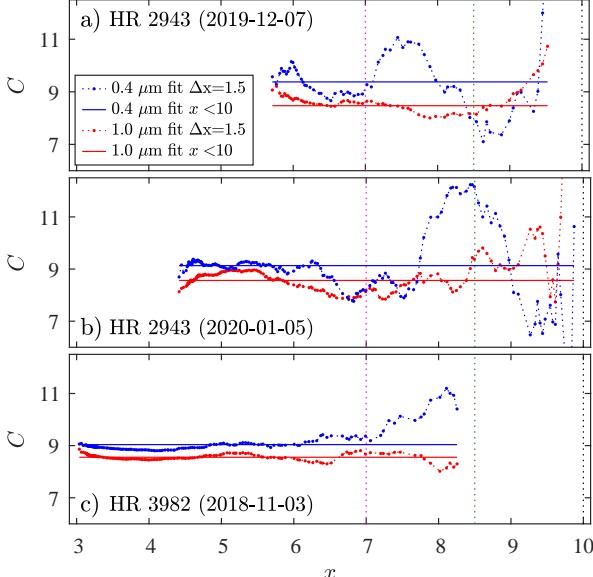

**Figure C1.** The variable curves show the calibration ($C$) variation for regressions associated with the low frequency (sliding window) curves of Figure 8. The horizontal lines correspond to the single $C$ value retrieved from the full (pan-$x$) regression lines of Figure 8, while the three colored (dotted) vertical lines correspond to the colors of the three $x_{max}$ cases of Figure 10.

## 265 Appendix C: Error discussion supplement

The $C$ values, derived from tangents applied to the Figure 8 solid curves (the means of a $\Delta x = 1.5$ sliding window), are plotted in Figure C1. The objective of this plot is to highlight more robust (lower frequency) C variations (and thus C errors) as a function of $x$. The 400 nm $C$ values are relatively stable up to $x \simeq 7$ to 7.5 where they are subject to a large increase. The 1000 nm $C$ pattern is similar with an increase beginning at $\simeq 9$ (observations that are roughly consistent with the vignetting 270 arguments of subsection 6.2).

*Author contributions.* Liviu Ivănescu: conceptualization, methodology, data curation, software, formal analysis, investigation, writing original draft. Norman T. O'Neill: validation, writing, review and editing, supervision, funding acquisition.

*Competing interests.* No competing interests are present.



*Acknowledgements.* This work was supported by CANDAC (the Canadian Network for the Detection of Atmospheric Change) via the

NSERC PAHA (Probing the Atmosphere of the High Arctic) project, by the NSERC CREATE Training Program in Arctic Atmospheric Science, as well as by the grant 21SUASACOA and the FAST program of the Canadian Space Agency (CSA). Finally we also gratefully acknowledge the support of the CANDAC operations staff at Eureka for their numerous troubleshooting interventions.



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
