# Peer review of "Multi-star calibration in starphotometry"

_EGUsphere, 2023_

## Author Response (AR1)

*The referee's comments are presented in italic* and our answers are written in plain text. **Modifications of the manuscript, if any, are written in plain bold text.**

**Response to Referee Comment #1**

We appreciate the holistic and clearly experienced overview of the paper, as well as the attentiveness with which the reviewer must have applied when reading through the paper.

**Response to Referee Comment #2**

We thank the referee for the in-depth and careful evaluation. The specific actionable points that were brought up (in italics) are addressed below. Please note that line numbers are in reference to the reviewed manuscript.

1. *"If the reader is not familiar with star photometry and previous publications by the authors, following the derivations of Eqs. 1 and 2 can be difficult. Just a very brief explanation can help."*

The reviewer will appreciate that the intent of this multi-star calibration paper is to follow on the heels of the comprehensive "Accuracy in starphotometry" paper (Ivanescu et al., 2021): in citing that paper we are also indicating where the reader can find the type of fundamental explanation that the reviewer is asking for.

To make that point clearer, we changed the lines 34-35 from: "(see Ivanescu et al. (2021) for the nomenclature details)" to "(see Ivanescu et al. (2021) for **a more comprehensive elaboration of this section**)."

We also made the following changes (lines 35-36): "Their corresponding (instrument) signals, expressed in terms of magnitude (logarithmic) formulation, are S_0 and S, respectively." was changed to "Their corresponding instrument signals, expressed in terms of magnitude, are S_0 = **-2.5 log F_0** and S = **-2.5 log F**, respectively, **with F_0 and F being the actual measurements in counts/s**."

2. *"The other point is about what are the star catalogues currently available, their spectral resolution and if they are public available. Again, a brief discussion could help."*

We appeal to the fact that catalogue information is also thoroughly described in Ivanescu et al., (2021). To better emphasize the availability of this information elsewhere, we changed line 63 from: "see Ivanescu et al. (2021) for a detailed discussion of error bias in the catalog stars" to "see Ivanescu et al. (2021) for a detailed discussion of error bias in the **Pulkovo and other** catalogues."

To specify our specific catalog, line 34 was changed from: "$M_0$ (provided by a catalog)" to "$M_0$ (provided by **the Pulkovo** catalogue **of Alekseeva et al. (1996)**)".

3. *"Finally, I believe that a brief explanation of the instrument used for acquiring the data is needed."*

We again appeal to the fact that our Eureka instrument was extensively described in Ivanescu et al. (2021). We complemented that information by changing the lines 17-19 from "Eureka, NU, Canada (Ivanescu (2015), Baibakov et al. (2015), Ivanescu et al. (2021))." to "Eureka, NU, Canada (Ivanescu (2015), Baibakov et al. (2015), Ivanescu et al. 2021)), **using a commercial spectrometer-based starphotometer[1], attached to a Celestron C11 telescope.**"

[1] **made by Dr. Schulz & Partner GmbH (currently closed).**

We also added a reference to our initial instrument development (Gröschke et al., 2009) in a general sentence about historical starphotometer developments. The following sentence was therefore inserted after the 1st sentence of the Introduction: "**Dedicated instrument development had already begun in the late 1950s (Dachs 1960; Dachs et al., 1966, Dachs, 1966), with increased activity after 2000 (Theoret, 2003; Gröschke et al., 2009; Perez-Ramirez, 2010; Oh 2015}.**"